# The Structural and Functional Responses of Rhizosphere Bacteria to Biodegradable Microplastics in the Presence of Biofertilizers

**DOI:** 10.3390/plants13182627

**Published:** 2024-09-20

**Authors:** Xueyu Cheng, Xinyang Li, Zhonghua Cai, Zongkang Wang, Jin Zhou

**Affiliations:** 1Marine Ecology and Human Factors Assessment Technical Innovation Center of Natural Resources Ministry, Tsinghua Shenzhen International Graduate School, Shenzhen 518055, China; chengxueyu1999@163.com (X.C.); lixinyan21@mails.tsinghua.edu.cn (X.L.); caizh@sz.tsinghua.edu.cn (Z.C.); 2Shenzhen Public Platform for Screening and Application of Marine Microbial Resources, Institute for Ocean Engineering, Shenzhen International Graduate School, Tsinghua University, Shenzhen 518055, China; 3Shenzhen Key Laboratory of Advanced Technology for Marine Ecology, Institute for Ocean Engineering, Shenzhen International Graduate School, Tsinghua University, Shenzhen 518055, China; 4Ecological Fertilizer Research Institute, Shenzhen Batian Ecological Engineering Co., Ltd., Shenzhen 518057, China

**Keywords:** biofertilizers, carbon metabolic potential, metagenomics, microplastics, rhizosphere bacteria

## Abstract

Biodegradable microplastics (Bio-MPs) are a hot topic in soil research due to their potential to replace conventional microplastics. Biofertilizers are viewed as an alternative to inorganic fertilizers in agriculture due to their potential to enhance crop yields and food safety. The use of both can have direct and indirect effects on rhizosphere microorganisms. However, the influence of the coexistence of “Bio-MPs and biofertilizers” on rhizosphere microbial characteristics remains unclear. We investigated the effects of coexisting biofertilizers and Bio-MPs on the structure, function, and especially the carbon metabolic properties of crop rhizosphere bacteria, using a pot experiment in which polyethylene microplastics (PE-MPs) were used as a reference. The results showed that the existence of both microplastics (MPs) changed the physicochemical properties of the rhizosphere soil. Exposure to MPs also remarkably changed the composition and diversity of rhizosphere bacteria. The network was more complex in the Bio-MPs group. Additionally, metagenomic analyses showed that PE-MPs mainly affected microbial vitamin metabolism. Bio-MPs primarily changed the pathways related to carbon metabolism, such as causing declined carbon fixation/degradation and inhibition of methanogenesis. After partial least squares path model (PLS-PM) analysis, we observed that both materials influenced the rhizosphere environment through the bacterial communities and functions. Despite the degradability of Bio-MPs, our findings confirmed that the coexistence of biofertilizers and Bio-MPs affected the fertility of the rhizosphere. Regardless of the type of plastic, its use in soil requires an objective and scientifically grounded approach.

## 1. Introduction

Traditional agricultural practices have long relied on inorganic fertilizers to manage soil and increase crop yields. However, the use of these fertilizers has resulted in various negative consequences, including increased soil salinity, reduced soil fertility, an exacerbation of the greenhouse effect, and contribution to water body eutrophication [1]. In response to these challenges, alternative methods that rely on organic inputs to enhance nutrient availability and safeguard the environment have been advocated in recent decades. Biofertilizers are viewed as an alternative to inorganic fertilizers in agriculture due to their potential to enhance crop yields and ensure food safety [2]. These organic products contain specific microorganisms [3] capable of injecting various nutrients into the soil by fixing nitrogen, dissolving or mineralizing phosphate and potassium, releasing plant growth regulators, and producing antibiotics [3]. Serving as microbial inoculants, biofertilizers enhance soil physicochemical properties, microbial community diversity, and plant growth [3]. Useful microbial populations in agriculture include plant growth-promoting rhizobacteria (PGPR), N_2_-fixing cyanobacteria, plant disease-suppressive bacteria, oil-toxicant degrading microbes, and actinomycetes [4,5].

Environmental contamination by microplastics (MPs) is now considered an emerging threat to soil ecosystems, and agricultural lands act as potential sinks for MPs [6]. Recent reports indicate that microplastic fragments can make up up to 7% of soil by dry weight [7]. Estimates suggest that 43–63 and 30–44 million tons of MPs are introduced annually to European and North American farmlands, respectively [8]. This microplastic contamination in agroecosystems often originates from farm practices, especially when using plastic mulches. Among these mulches, the main production materials are durable polymers, such as polyethylene (PE) and polypropylene (PP), which are not biodegradable and can persist in the environment for decades [9]. Some biodegradable mulches have been developed to achieve greener and more environmentally friendly plastic mulches [10,11]. As an alternative to traditional agricultural mulches, biodegradable polymers such as polylactic acid (PLA) and polybutylene terephthalate (PBAT) are increasingly being used [12]. However, the degradation of many biodegradable polymers under environmental conditions can be slow or incomplete, leading to the accumulation of Bio-MPs in soil [13].

The rhizosphere, which refers to the narrow zone surrounding and influenced by plant roots, is a hotspot for numerous organisms and is considered as one of the most complex ecosystems on Earth [14]. Each gram of root can contain up to 10 [11] rhizosphere microorganisms [15]. Rhizosphere microorganisms can promote plant growth through various mechanisms, including nitrogen fixation [16], promoting the uptake of micronutrients such as Fe [17], releasing cations through mineral weathering [18]. They can also produce antibiotics [14], and regulate the plant’s immune system to protect the plant from pathogen attacks [19]. The coexistence of biofertilizers and MPs is becoming increasingly common due to their parallel appearance in the environment [1]. And the use of both can have direct and indirect effects on rhizosphere microorganisms. Previous research has primarily focused on studying the effects of soil physicochemical properties, soil enzyme activities, and plant growth and development in the presence of biofertilizers or MPs alone. Although some studies have examined the effects on soil microorganisms, few have specifically investigated the rhizosphere microenvironment [20,21,22,23,24]. Therefore, further studies are necessary to investigate the responses of rhizosphere microbial communities in the coexistence of biofertilizers and MPs and enhance our understanding of the impact on the rhizosphere environment. For conventional MPs, MPs can change the features of soil (e.g., pH value, water evaporation, and aggregate stability) [5,25], microbial profiles (e.g., diversity, composition, and organic matter metabolic ability) [26], and the physicochemical properties of crops (e.g., biomass, growth, and root development) [2]. The interaction between Bio-MPs and biofertilizers and their impact on soil microecological characteristics are relatively unknown compared with conventional plastic particles. Little is known about the effects of Bio-MPs (as regards coexistence with biofertilizer) on soil microecological processes, especially the rhizospheric bacterial response.

In this study, we investigated the responses of rhizosphere microbial communities in the coexistence of biofertilizers and Bio-MPs and enhanced our understanding of the direct or indirect impact on the rhizosphere. In addition, biodegradable plastics exhibit a different picture of rhizosphere microorganisms than traditional plastics. We used *Bacillus amyloliquefaciens* biofertilizer as an example, and two different types of MPs, PE-MPs (the most typical staple material for agricultural films and packaging materials) and Bio-MPs (poly (butylene adipate-co-butylene terephthalate)-poly(lactic acid) (PBAT-PLA) blends, an increasingly popular biodegradable plastic) as the target materials to investigate. We investigated the effects of MPs on the rhizosphere microorganisms of *Brassica parachinensis* in the context of biofertilizer application. Specifically, we (1) analyzed the changes in the bacterial diversity, composition, and network complexity of the rhizosphere microbial community using 16S rRNA sequencing and (2) analyzed the changes in the functioning (especially the carbon cycle potential) of the rhizosphere microbial community using metagenomic tools. The ultimate goal was to evaluate the impact of degradable plastic particles on the soil microecological environment.

## 2. Results

### 2.1. Changes in the Physicochemical Properties of the Rhizosphere

The inclusion of 0.2% and 2.0% Bio-MPs led to a significant increase in SOC (96.87% and 176.81%) and OM (90.60% and 158.03%), which also showed dose dependence (Figure 1a,b). By comparison, PE-MPs increased SOC (27.57%) and OM (20.15%) at only high concentrations (2.0%), and the effects were significantly weaker than those of Bio-MPs (Figure 1a,b). Additionally, the presence of Bio-MPs also caused a significant increase in the pH of the rhizosphere soil, changing it from weakly acidic (6.79) to weakly alkaline (7.92–8.11) compared to the BF group (Figure 1c). For other physicochemical parameters (TN, TP, TK, AHN, and AK), there were slight variations to Bio-MPs and exposed concentration (Appendix A). Both MP materials significantly decreased the AP level compared with the blank group. The decrease was approximately one of 3.5–4 times (Appendix A).

### 2.2. Changes in Bacterial Communities in the Rhizospheric Environment

The raw sequences were subjected to quality control and sequence screening before being optimized based on the 97% similarity threshold. This resulted in 18,485 bacterial operational taxonomic units (Appendix A). The richness and diversity of the communities in the different treatment groups were assessed using the ACE and Shannon indexes. Compared to the BF group, the ACE index of bacteria of the PE-H group significantly reduced (Figure 2a). Both treatment groups of Bio-MPs also significantly reduced the alpha-diversity of bacteria and followed the concentration-dependent effect (Figure 2a,b). For β-diversity, PERMANOVA revealed significant differences in bacterial community composition structure among the six treatment groups (F = 4.584, *p* < 0.01) (Figure 2c).

As for the bacterial composition, the community in the rhizosphere varied significantly depending on the treatments. Among all the groups, Proteobacteria was the dominant taxa at the phyla level, followed by Actinobacteria, Firmicutes, and Bacteroidetes (Figure 2d).

A linear discriminant analysis effect size (LEfSe) was performed to clarify the differences among the treatment groups. A total of 83 biomarkers were identified (Appendix A). The six groups of CK, BF, PE-L, PE-H, Bio-L, and Bio-H had 11, 25, 6, 8, 6, and 27 biomarkers, respectively. The BF group showed significant enrichment of Acidobacteria, Nitrospirae, and Firmicutes. After MP exposure, the indicators were changed. The indicators in the PE groups were Janthinobacterium, Bacillus, and Parasegitibacter. Similarly, the biomarkers in the Bio groups were Comamonas, Cupriavidus, and Burkholderia.

We found that adding plastic particles changed the sensitivity of rhizosphere microorganisms to environmental parameters after conducting a redundancy analysis (Appendix A). The Bio groups were positively (significantly) correlated with pH, SOC, OM, and TK. However, these factors were negatively (significantly) correlated with the bacterial community in the CK, BF, and PE groups. Meanwhile, AP, AHN, and AK showed a significant negative correlation with Bio groups and a significant positive correlation with CK, BF, and PE groups (Appendix A). More detailed results of the relative abundance of key bacterial taxa and soil environmental factors are displayed in the heat maps (Appendix A).

### 2.3. The Network Profiles of the Rhizosphere Bacterial Community

The co-occurrence model of different treatments (100 high-ranking bacteria at the genus level) was analyzed to reveal rhizosphere bacterial community responses to biofertilizers and MPs. The main topological parameters include edges, nodes, the percentage of positive/negative edges, average path length, graphic density, network diameter, average degree, and modularity, as summarized in Figure 3. Overall, the percentage of positive edges (56.25–86.07%) outnumbered the negative edges in all test groups (13.93–43.75%) (Figure 3), indicating that bacterial connection types tended to be positively correlated. The graphic density and the number of edges in the BF group decreased by 38.2% and 35.9%, respectively, compared to group CK. The percentage of positive edges decreased by 19.4%, while the percentage of negative edges increased by 120%. These findings suggest that the addition of biofertilizers weakened the linkage of the rhizosphere bacterial community, resulting in low network complexity. Additionally, the average degree and clustering coefficient decreased by 37.2% and 2.0%, respectively, as the network aggregation decreased (Figure 3). PE-H and Bio-H had opposite effects compared to BF. The graph density, number of edges, average degree, and clustering coefficient of PE-H decreased by 12.5%, 14.4%, 13.5%, and 1.3%, respectively. In contrast, the parameters mentioned above increased in the Bio-H group by 106.8%, 112.8%, 110.7%, and 19.4%, respectively (Figure 3). These results suggest that the presence of 2.0% PE-MPs reduces the complexity and density of the network, whereas 2.0% Bio-MPs enhances it.

### 2.4. Rhizosphere Bacterial Function Based on Metagenomic Analysis

Comparative metagenomic analyses were conducted on the CK, BF, PE-H, and Bio-H groups to determine the effects of MPs present with biofertilizers on rhizosphere bacterial function. The gene sets from each treatment group were annotated using the KEGG primary and secondary functional classification pathways (Appendix A). Figure 4a demonstrates a reduction in the relative abundance of most secondary functions in the Bio-H group, with a stronger reduction than in PE-H. These functions include “amino acid metabolism”, “carbohydrate metabolism”, “lipid metabolism”, and “energy metabolism”. In contrast, “cell motility”, “cell growth and death”, “cellular community and signal transduction”, and “membrane transport” were upregulated (Figure 4a). The clustering results indicated that the BF and PE-H groups had similar features. However, treated Bio-H and PE-H showed the greatest difference in functional composition (Figure 4a).

In addition to the common core functions, representative key functions (LDA score > 2.5) for each group were also determined using LEfSe analysis. Within the KEGG tertiary functional classification pathways, the BF group was enriched in “oxidative phosphorylation” and “metabolic pathways”, with only “fatty acid metabolism” enriched in PE-H. Similarly, the Bio-H group showed enrichment in four pathways: “fatty acid degradation”, “benzoate degradation”, “valine-leucine and isoleucine degradation”, and “two-component system” (Figure 4b).

Considering the metabolic potential of rhizosphere microorganisms under the influence of plastic particles (PE-MPs and Bio-MPs), these enriched functional genes are closely related to carbon metabolism. Therefore, we researched carbon cycle genes in the rhizosphere environment further. Fifteen pathways related to carbon cycle genes (e.g., carbon fixation, central metabolism, and methane metabolism) were identified in the rhizosphere across all treatment groups (Appendix A). A total of 184 genes (KO numbers) are involved in the metabolic pathways of the carbon cycle, with some KO numbers being associated with multiple pathways. The tricarboxylic acid cycle was the predominant pathway, constituting 17.39% of the overall carbon metabolism function (Appendix A).

Figure 4c shows that the distribution of genes related to the carbon cycle varies across different groups. Biofertilizers partially downregulated the expression of carbon cycle genes. The introduction of MPs also results in alterations to the carbon cycle. The expression of carbon cycle genes was upregulated in PE-H. At the same time, many genes were further downregulated in Bio-H (Figure 4c). The pathways of carbon fixation and central metabolism did not show significant differences among the four groups (Figure 4d). However, the presence of biofertilizers and MPs had a significant impact on the pathways of methane metabolism. Compared to the CK group, the BF and PE-H groups significantly increased methane production, whereas BF-Bio20 had a significant inhibitory effect on the conversion of methane to methanol (Figure 4d).

A partial (geographic distance-corrected) Mantel test was performed to explore the correlation between carbon cycle genes and environmental factors and to clarify the contribution of environmental factors to carbon metabolism (Figure 5). TK had a significantly strong correlation with functional genes related to methane metabolism in terms of taxonomic and functional composition (*p* < 0.05). Significant positive correlations were also observed between TK-TN, pH-SOC, pH-OM, SOC-OM, and AK-AHN. Bacterial species also contributed to environmental parameters related to carbon metabolism. Appendix A shows that multiple species of Acidovorax, Nitrospira, Nocardioides, Noviherbaspirillum, Paraburkholderia, Phenylobacterium, Pseudolabrys, Trinickia, and Massilia contribute to at least one carbon cycle pathway. Devosia was found exclusively in the central metabolism of the CK and BF groups. Opitutus was identified only in the central metabolism of the BF group, while Methylobacillus was identified in the methane metabolism of the CK group.

### 2.5. Partial Least Squares Path Model (PLS-PM) Analysis

PLS-PM analysis was conducted to investigate the potential relationships between biofertilizers and MPs, soil physicochemical properties, bacterial diversity, and related carbon cycling potential. The results (Figure 6) indicate that biofertilizers and MPs had a direct and significant positive correlation with soil pH, SOC, and OM and a negative correlation with soil nutrient content and bacterial diversity, affecting rhizosphere carbon cycling. Simultaneously, the addition of biofertilizers and MPs has a direct impact on the carbon cycle in the rhizosphere. In the natural environment, the coexistence of biofertilizers and MPs will affect the physical and chemical properties of soil bacterial activities in the rhizosphere and drive carbon turnover in the rhizosphere environment.

## 3. Materials and Methods

### 3.1. Preparation of Biofertilizers and Characterization of MPs

The *B. amyloliquefaciens* biofertilizer was provided by the Shenzhen Batian Ecological Science and Technology Co., Ltd., Shenzhen, China [27]. The PE-MPs and PBAT-PLA Bio-MPs were purchased from Huachuang Chemicals Co., Ltd., Dongguan, China. Scanning electron microscopy (Phenom ProX, Eindhoven, The Netherlands), Fourier transform infrared spectroscopy (FT-IR) (Thermo Scientific Nicolet iS-50, Waltham, MA, USA), and laser particle size analysis (Malvern Mastersizer 2000, Malvern, PA, USA) were used to determine the morphology, composition, and particle size of the two types of MPs (Appendix A). The production process of the biofertilizer and information on the MPs can be found in Appendix A.

### 3.2. Exposure Experiment of MPs

A 28-day greenhouse pot experiment was conducted from 28 November to 26 December 2022, at the Shenzhen Batian Ecological Fertilizer Research Center in Guangdong Province, China. The test soil came from farmland in Guangming District that had not been treated with plastic mulch. Appendix A shows the physicochemical properties of the soil at the site.

The biofertilizers used in our potting experiments were referenced in our previous work (Wang et al. [22]). Each pot contained three 14-day-old *B. parachinensis* plants with the same growth. We used two concentrations of MPs (0.2% and 2.0% *w*/*w*, dry weight ratio of MPs to soil), with references from previous studies [25], to simulate MP concentrations in the real environment. The two concentrations represented low- and high-strength mulch use, respectively.

Six treatment groups were designed, including group CK (an untreated control with no biofertilizer or MPs added), group BF (with biofertilizer only), groups PE-L/PE-H (with biofertilizer and 0.2%/2.0% of PE-MPs), and groups Bio-L/Bio-H (with biofertilizer and 0.2%/2.0% of Bio-MPs). Each treatment group comprised six biological replicates (pots) containing 4 kg of test soil and three *B. parachinensis* seedlings.

### 3.3. Physicochemical Properties of Rhizosphere Soil

Rhizosphere soil sampling is described in Appendix A. The physicochemical properties of the rhizosphere soil were determined as follows: pH was determined using a potentiometric method with a soil/water ratio of 1:2.5 *w*/*v*. Soil organic carbon (SOC) and organic matter (OM) contents were determined using the potassium dichromate oxidation–external heating method [28]. Total nitrogen (TN) content was determined using the Kjeldahl method. The study analyzed the contents of total phosphorus (TP), total potassium (TK), and alkaline hydrolysis nitrogen (AHN) using molybdenum blue colorimetry, flame photometry, and the alkali diffusion method, respectively. The available phosphorus (AP) content was determined using the molybdenum-antimony inverse colorimetric method after soaking the inter-root soil in ammonium hydrogen fluoride and sodium bicarbonate solutions [28]. The available potassium (AK) content was determined using ammonium acetate leaching and flame photometric methods [29].

### 3.4. DNA Extraction, PCR Amplification, and Metagenomic Analysis

The bacterial communities in the rhizosphere soil of the six treatment groups were characterized using high-throughput sequencing. DNA was extracted from the rhizosphere soil using cetyltrimethylammonium bromide (CTAB). DNA purity and concentration were assessed using 1% agarose gel electrophoresis. The gene amplification of different regions (V3–V4) of bacterial 16S rRNA was performed using primers 341F (CCTAYGGGRBGCASCAG) and 806R (GGACTACNNGGGGTATCTAAT). The DNA extraction process and assessment of microbial diversity are described in Appendix A.

Metagenomic sequencing analysis was performed for bacterial function. The CK, BF, PE-H, and Bio-H groups were collected as targets. Metagenomic sequencing was performed using the NEBNext@ UltraTM DNA Library Prep Kit for Illumina (New England Biolabs, Massachusetts, USA) according to the manufacturer’s recommendations. Briefly, indexing codes were added to the sequences for each set of samples. DNA samples were sonicated to obtain 350 bp fragments, which were then end-polished, a-tailed, and ligated to full-length aptamers. This process was followed by PCR amplification and purification of the PCR products using the AMPure XP system. Library size distribution was analyzed on an Agilent 2100 Bioanalyzer and quantified using real-time PCR. Index-coded samples were clustered on the cBot clustering system according to the manufacturer’s instructions, and the DNA libraries were finally sequenced on the Illumina Novaseq 6000 platform, generating 150 bp paired reads. Detailed sequence quality control and functional assembly information are provided in Appendix A.

### 3.5. Statistical Analyses

Statistical analyses were performed using IBM SPSS Statistics 25.0, specifically one-way analysis of variance (ANOVA) with a *p*-value ≤ 0.05 for significance, and implemented in SPSS 25 software (IBM, New York, NY, USA). All experimental parameters were measured at least in triplicate, whose results are presented as the mean ± standard deviation. Data were analyzed and graphically presented using R Studio (4.1.0) and OriginPro 2019. Two alpha diversity indices, comprising ACE and Simpson, were analyzed using the “vegan” and “picante” packages in R (v4.1.0). To further characterize the microbial community, we used OriginPro 2019 (OriginLab, Northampton, USA) software to draw histograms of the relative abundance of bacteria at the phylum level. Comparisons between multiple groups were achieved using the linear discriminant analysis effect size (LEfSe) method to identify biomarkers that differed significantly in abundance between groups (Kruskal–Wallis rank sum test, linear discriminant analysis (LDA) score > 3.5). In addition, redundancy analysis (RDA) was used to assess the effect of soil physicochemical properties on the microbial community, and its significance was determined by Monte Carlo permutation (permu = 999). We conducted a correlation heatmap analysis in R software to investigate the relationships between environmental factors and species composition at the phylum level. Spearman’s coefficient (|ρ| > 0.6) was used to assess the degree of correlation between species at the genus level, these visualized using Gephi 0.10 software. The Circos diagram shows the abundance of KEGG primary and secondary functional classification pathways. Partial Least Squares Path Modeling (PLS-PM) with the “plspm” package in R was used to explore the direct, indirect, and interactive effects of MPs, soil physicochemical characteristics, microbial community diversity, and carbon cycle functional gene abundance. The“psych”and“ggplot2”packages in R software were used for correlation analyses among different treatments.

## 4. Discussion

### 4.1. Bacterial Composition Responses to Biofertilizers and MPs

As expected, the addition of biofertilizer resulted in significantly high rhizosphere SOC and OM content, attributed to its 29.27% exogenous organic matter (Appendix A). After MP exposure, especially to Bio-MPs, the SOC and OM contents were further increased, likely due to their ability to release endogenous organic matter during degradation. Biodegradable plastics comprising 60–80% of carbon undergo breakdown in the natural environment [30]. Bio-MPs and their degradation products act as supplementary biologically effective carbon, directly affecting soil OM content. In addition, we found that both MP additions significantly increased the pH of the rhizosphere, changing it from weakly acidic to weakly alkaline. This increase in pH is consistent with the findings of Liu et al. [31]. The potential reason was that the addition of MPs led to the competitive adsorption of hydrophobic organic compounds with OM, which in turn altered the acid–base balance of the soil [30].

In addition to the nonbiological parameters (physicochemical factors) of the rhizosphere environment, exposure to MPs also affects the biological parameters of the rhizosphere. The inclusion of high concentrations of MPs, particularly in the Bio group, led to a significant decrease in the richness and diversity of the bacterial community compared to the sole BF group (Figure 2). We found that the negative effect of MPs was significantly greater than the promoting effect of BF when compared with the CK group. These findings are supported by previous research that showed the negative effects of MPs on rhizosphere bacterial diversity and richness [32]. Possible reasons include, as a hydrophobic polymer with a large specific surface area, how MPs can absorb multiple substances in the rhizosphere, which contributes to the enrichment of specific bacteria [30,32]. Additionally, some antioxidants, colorants, and plasticizers added to MPs are released into the environment and act as nutrient sources for certain bacteria [33]. Third, MPs may facilitate colonization by bacteria capable of degrading polymers as carbon sources. Although PE-MPs are less biodegradable and are unlikely to affect soil microorganisms directly, they can drive the selection of specific microbial taxa with potential degradation capabilities [34]. For instance, certain members of Actinobacteria can break down PE-MPs by producing hydrolases [35,36]. When compared to conventional PE-MPs, Bio-MPs decompose faster. In this case, Bio-MPs with rough surfaces release more carbon sources, which is conducive to bacterial colonization [30]. As a result, Bio-MPs have a greater impact on bacterial alpha-diversity. Meanwhile, the beta-diversity was also influenced by exposure to MPs and displayed concentration dependence (Figure 2b). Combined with Appendix A, we speculated that the dose and type of MP are co-drivers of bacterial community composition [37].

Further, significant differences in bacterial composition were found under different treatments, indicating that biofertilizers and MPs co-influence the structure of bacterial communities. Several studies have shown that Proteobacteria, Actinobacteria, Bacteroidetes, and Firmicutes play critical roles in organic mineralization and C, N, and S cycling [38,39]. The content of Proteobacteria increased slightly after biofertilizer application, with the highest content found in the Bio-H group. Previous research has identified Proteobacteria as the primary decomposer of OM [40]. As the main taxonomic unit in soil samples, Proteobacteria play an integral role in the carbon cycle and tend to colonize nutrient-rich environments [41]. Acidobacteria plays a crucial role in the carbon cycle and can process a variety of carbon sources [41]. However, the abundance of Acidobacteria was significantly reduced in the Bio treatment group. Acidobacteria is a typical acidophilic bacterium that thrives in acidic and oligotrophic ecological niches rather than nutrient-rich alkaline environments [42]. In this study, adding MPs increased the rhizosphere pH and nutrient content, especially for Bio-MPs. This finding provides supplementary evidence for the reduction in Acidobacteria. The LEfSe results also indicate that the exposure of MPs could screen their biomarker (Appendix A). Considering Bio groups as an example, *Comamonas* and *Burkholderia* were the main indicators in the Bio-L and Bio-H groups. *Comamonas* is a multifunctional bacterium capable of degrading phenols, PAHs, and heterocyclic aromatic hydrocarbons [43]. The enrichment of *Comamonas* may be related to the biodegradation of PBAT, which is a fully biodegradable aliphatic-aromatic co-polyester in Bio-MPs [44].

Network analysis has been widely used to explore co-occurrence patterns in soil microbial communities [45,46]. This study found that the addition of biofertilizers resulted in a decrease in network complexity and network aggregation, weakening the linkages of rhizosphere bacterial communities. These findings are inconsistent with previous research [47]. Furthermore, the percentage of negative edges increased while positive edges decreased, indicating that competition, rather than cooperation, was the prevailing factor in the bacterial community. The use of biofertilizers was inferred to introduce many beneficial bacteria into the rhizosphere. This condition led to overlapping dominant bacterial ecological niches [48], increasing competition, and decreasing linkages between rhizosphere bacterial communities. MP exposure further reduced the complexity and density of rhizosphere bacteria networks, which is consistent with the study of Wu et al. [49]. As organic pollutants, PE-MPs can impact the symbiotic pattern of rhizosphere bacterial communities. This situation can lead to the creation of synergistic bacterial symbiotic modules that promote the degradation of PE-MPs. Bio-MPs significantly impacted the total number of links in the rhizosphere bacterial network, particularly the percentage of positive edges, facilitating cooperation among bacteria. This outcome may be attributed to the degradable nature of Bio-MPs, resulting in the rhizosphere becoming a resource-rich environment. As a result, bacteria exhibit swift responses to the nutrients in the rhizosphere [35]. Based on these observations, we propose that Bio-MPs can promote the formation of complex bacterial community networks in the rhizosphere.

### 4.2. Rhizosphere Bacterial Function Responses to Biofertilizers and MPs

While the higher sequencing depth of metagenomic sequencing can determine what the altered microbes are doing, 16S rRNA sequencing can explain how the microbial structure is altered in rooted environments. The breakdown of biofertilizers and MPs by microbes is a long-term biometabolic process involving OM. Therefore, rhizosphere bacteria were annotated with the largest “Metabolism” genes. Our work found that the genes identified were primarily associated with metabolic processes that support bacterial activity, which facilitates nutrient cycling by rhizosphere microbes. Using Bio-MPs resulted in a decrease in metabolic processes and an increase in “cellular processes” and “environmental information processing”. This result suggests that adding Bio-MPs leads to changes in bacterial rhizospheric function.

Considering that carbon is one of the most important elements for plant growth, and through differential metabolic gene analysis, we found that the functional pathway mainly involves carbon metabolism processes. Therefore, we focused on carbon and analyzed its metabolic differences in various groups (Figure 4a,b). The use of biofertilizers induced alterations in carbon cycle genes. After adding MPs, different types had varying effects on rhizosphere bacterial functions. Ma et al. found that a moderate amount of MPs promoted nutrient cycling, but exceeding the threshold concentration, which harmed soil nutrient cycling [50]. Our study found that adding 2.0% PE-MPs upregulated the expression of carbon cycle-related genes, promoting nutrient cycling (Figure 4b). These results are consistent with the study by Ma et al. [50]. However, 2.0% of Bio-MPs suppressed most functional genes related to carbon cycling. The threshold of Bio-MPs is low, meaning they can adversely affect carbon cycling at low concentrations. Upon further evaluation of the relative abundance of the various pathways of the carbon cycle, Bio-MPs had a significant inhibitory effect on methane oxidation (Figure 4d). This inhibition could partially explain the lack of activity observed in *Methylobacillus*, which utilizes methane oxidation-derived carbon as a nutrient in the Bio group (Appendix A). The rhizosphere SOC and OM contents were significantly higher in the BF and PE-H groups. This increase may promote the growth of methanogenic bacteria [51], leading to increased methane production (Figure 4d). Ma et al. found a significant correlation between environmental factors, such as SOM and TN, and the abundance of functional genes related to methane metabolism in the rhizosphere after exposure to MPs [50]. This study conducted partial Mantel tests and found a significant and strong correlation between TK and methane metabolism-related functional genes (Figure 5), suggesting that methane metabolism is vulnerable to changes in environmental factors when biofertilizers and MPs coexist.

Changes in epigenetic function are responsible for the composition of microorganisms. We found that different bacterial species responded to different functions after calculating the relative abundance of bacteria involved in the three pathways (Appendix A). For instance, *Nitrospira* and *Paraburkholderia* were enriched in carbon fixation in the PE-H and central metabolism in the Bio-H group, respectively. *Paraburkholderia* is a bacterium capable of C–N fixation [52], while *Nitrospira* plays an important role in the N–C coupling cycling [53]. *Trinickia*, *Mitsuaria*, and *Phenylobacterium* were relatively enriched in carbon fixation and central metabolism, when treated with Bio-MPs. *Trinickia* is a PGPR [54], *Mitsuaria* can reduce the growth of pathogens and act as a biocontrol agent [55], and *Phenylobacterium* is associated with the degradation of PAHs [56]. These results indicated that the response of bacteria to MPs is typically nonlinear and microbes have structural variability and functional plasticity.

## 5. Conclusions

This study shows that the coexistence of biofertilizers and microplastics significantly impacts the rhizosphere environment and related bacteria, ultimately affecting the soil carbon cycle. Our work confirmed that MP exposure (i) altered the physicochemical parameters of rhizosphere soil and the bacterial composition, in particular, of the OM decomposing taxa; (ii) decreased the co-occurrence complex and the buffer ability to environmental stresses; and (iii) affected the relative abundance of carbon cycle bacteria, which in turn affects methane metabolism. PLS-PM analyses showed that adding biofertilizers and MPs improved SOC, OM, and the nutrients of soil but reduced its bacterial diversity and carbon cycling ability. Additionally, the linkage analysis shows that the ecological ramifications of exposure to Bio-MPs are bigger than conventional MPs for the rhizosphere of *B. parachinensis*. The results provide insight into the impact of biofertilizers and MPs on the rhizosphere microenvironment when they coexist. They warn us that although biodegradable plastics are a valuable alternative product, their negative effects in actual soil environments are worth paying attention to.

## Figures and Tables

**Figure 1 plants-13-02627-f001:**
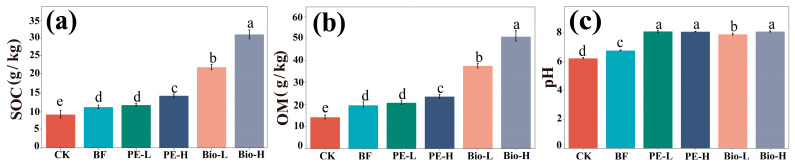
The effect of MPs on soil property existence with biofertilizer: (**a**) SOC; (**b**) OM; (**c**) pH. Data are presented as the mean ± SD (*n* = 6). Different letters (a, b, and c) were statistically different (*p* < 0.05) according to Duncan’s new multivariate range test.

**Figure 2 plants-13-02627-f002:**
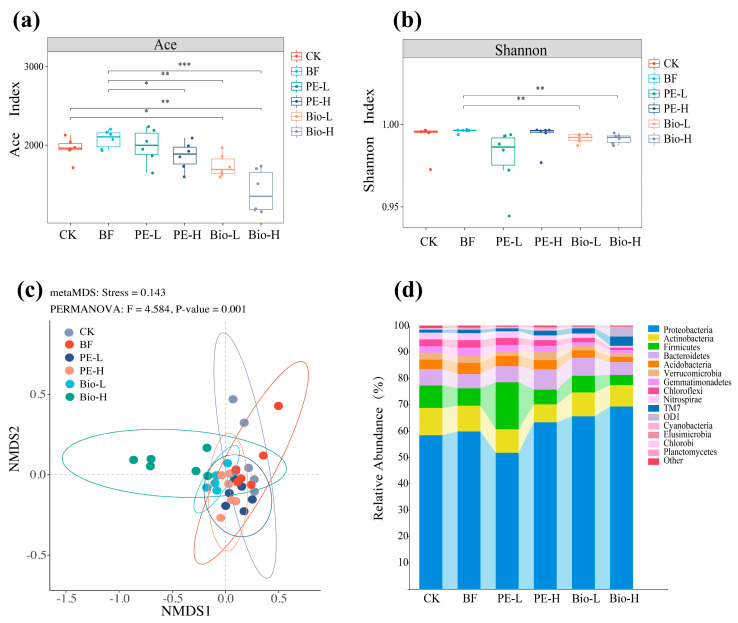
The effects of biofertilizers and MPs on rhizosphere bacterial community structure and composition. (**a**) ACE and (**b**) Shannon index. Statistical significance was indicated by *, **, and *** for *p* < 0.05, *p* < 0.01, and *p* < 0.001, respectively. (**c**) Non-metric multidimensional scaling (NMDS) plots with corresponding pressure values for different bacterial groups at the OTU level. (**d**) Relative abundance maps of bacterial distribution at the phylum levels.

**Figure 3 plants-13-02627-f003:**
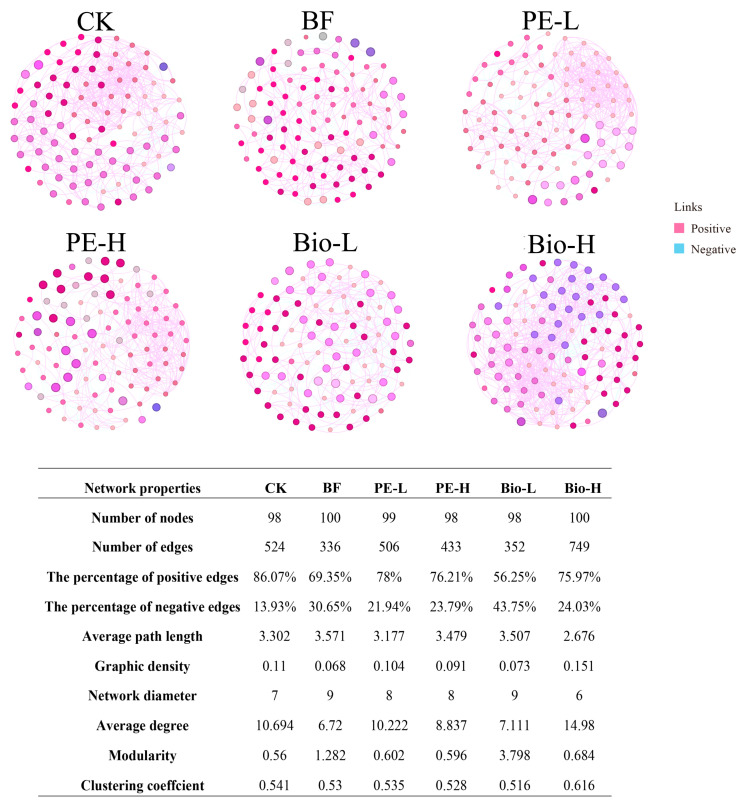
The effect of biofertilizers and MPs on the network of rhizosphere bacterial communities at the genus level. Nodes are colored according to module, with pink edges indicating a positive correlation and blue indicating a negative correlation. Connections are defined as strongly correlated (Spearman’s |*ρ*| > 0.6) and significantly correlated (*p* < 0.05). The topological parameters of microbial networks obtained in all treatments are shown in the table.

**Figure 4 plants-13-02627-f004:**
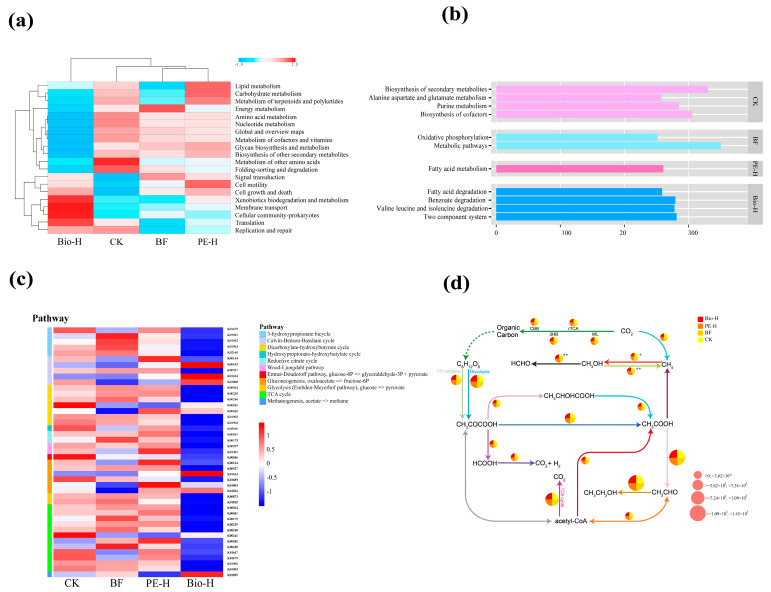
The effects of biofertilizer and MPs on the function and pathways related to carbon cycling in rhizosphere bacteria. (**a**) A heatmap of KEGG secondary functional clustering for the top 20 abundances in different treatment groups. Hierarchical clustering (Bray–Curtis) analysis of all samples based on the similarity of KEGG pathways; (**b**) LEfSe analysis of different treatment groups under KEGG tertiary levels of functional annotation. (**c**) A heat map of carbon cycle genes, and (**d**) the relative abundance of the carbon cycle pathway. The pie chart shows the relative abundance of each pathway for each sample clock, and the size of the pie chart shows the total relative abundance of that pathway. The * sign in the upper-right corner of the pie chart indicates the abundance of the pathway differs significantly between different groups of samples. The * and ** indicate significance at *p* < 0.05 and *p* < 0.01.

**Figure 5 plants-13-02627-f005:**
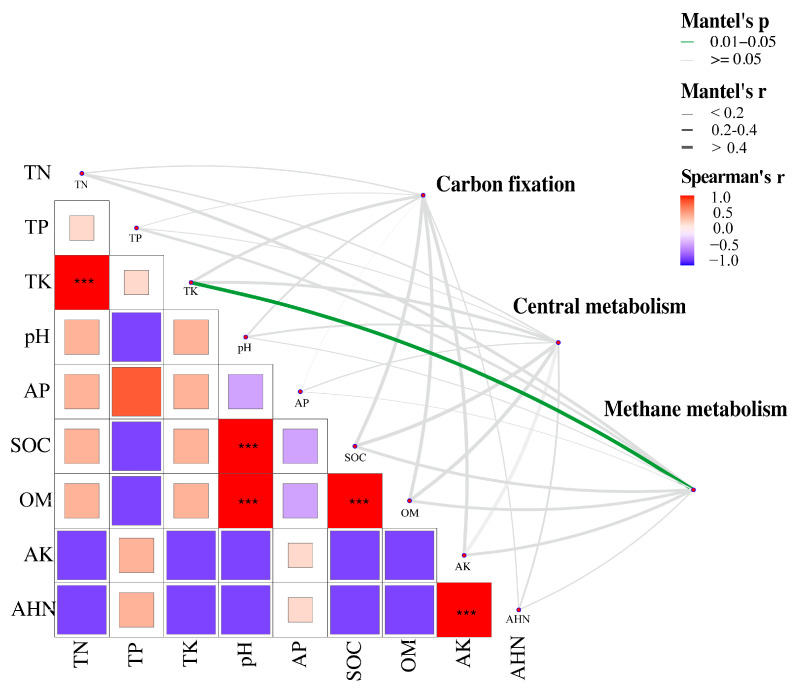
A partial Mantel test between functional genes and environmental factors under biofertilizer and MP exposure. The statistical significance based on 9999 permutations is indicated by the edge color, while the Mantel r statistic correlation for the corresponding distance is represented by the edge width. The Spearman correlation coefficient is shown by the color gradient. *** denotes that the significance level for the Spearman correlation coefficient is 1.

**Figure 6 plants-13-02627-f006:**
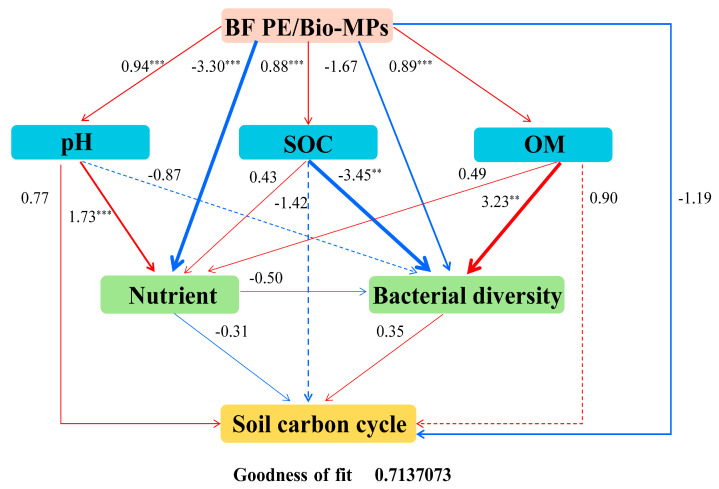
PLS-PM analysis showed the relationship between MP type, soil physicochemical properties, bacterial diversity, and rhizosphere carbon cycling (goodness of fit = 0.713). The width of the arrow indicates the strength of the path factor. The red and blue lines represent positive and negative effects, respectively. Solid and dashed lines indicate direct and indirect effects, respectively. In the PLS-PM model, the number on the line represents the total effect value (statistical significance was denoted by ** and *** for *p*-values less than 0.01 and 0.001).

## Data Availability

Sequence data used in this study were submitted to the Sequence Read Archive of the National Center for Biotechnology Information (accession number: 16S rRNA: RRJNA1039250; Macrogenome: PRJNA1080489).

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
