# Peer review of "The Structural and Functional Responses of Rhizosphere Bacteria to Biodegradable Microplastics in the Presence of Biofertilizers"

_plants, 2024, doi:10.3390/plants13182627_

Round 1

Reviewer 1 Report

Comments and Suggestions for Authors

The manuscript “Structural and functional responses of rhizosphere bacteria to

biodegradable microplastics in the presence of biofertilizers” by Cheng et al. demonstrates the effects of coexisting biofertilizers and Bio-Mps on the structure and functions of rhizosphere bacterial communities. They used amplicon sequencing and metagenome to investigate the rhizosphere microbial composition and functions under the conditions of the coexisting biofertilizers and Mps. Here, I have some comments for this manuscript:

1. Line 21-23: The first sentence revealed the importance of Bio-Mps, but the second sentence directly mentioned that the influence of coexistence of “Bio-Mps and biofertilizers” remains unclear. There is still a gap that why the authors studied the coexistence of “Bio-Mps and biofertilizers”.

2. Line 24: Function also includes carbon metabolic.

3. Line 35: Did the authors analyze the niche in this manuscript?

4. Line 78-80: Also, there is a gap why studied rhizosphere. 

5. Line 85: Is there any content about “slowing microbial activities” in this manuscript?

6. I am confused why the authors used so many different taxonomy levels for analyses, OTU level (?) for alpha diversity, family level for beta diversity, phylum and order level for composition,  genus level for LEfSe and network.

7. Did the authors normalized the data for 16S and metaG?

8. Line 122: There is no legend for Fig. 2b.

9. Line 450: What does “ functional assembly” mean?

10. Line 455: R Studio is just an IDE. 

11. Fig.5: Please check if Mantel’s r could be negative. And I think there is no need to show the line which Mantel’s p > 0.05.

12. Line 236-238: Please check if the genus name is in italic. 

13. Fig. 6: It seems that there is no any significant relationship between soil carbon cycle and other factors. It is wired because the soil properties changed a lot, especially SOC and OM. So the carbon cycle should be affected, just like what showed in Fig. 4c. There is a possibility that it is not proper to use carbon cycle gene as a index, since it combined a lot of different pathways.

Comments on the Quality of English Language

As for the language, the manuscript is a little bit hard to read. I suggest the authors to reorganize the manuscript and simplify the results descripition for better clarity and readability. For example, move M&M after introduction or conclusion. If after conclusion, please add the whole name of the abbreviations for their first appearance. Besides, please add the detailed information of bioinformatic part in methods. In somehow, the method in SI is important than in M&M. Additionally, please check some unprofessional expression like “the picture was changed” in Line 210 and “bipartite reads” in Line 450.

Reviewer 2 Report

Comments and Suggestions for Authors

The article entitled "Structural and Functional Responses of Rhizosphere Bacteria to Biodegradable Microplastics in the Presence of Biofertilizers" explores the combined effects of biofertilizers with both bioplastics and polyethylene plastics on the soil microbiome. The impact of biofertilizers on the rhizosphere microbiome has been extensively studied in the past. However, the recent rise in the use of bioplastics in agriculture, particularly for mulching, has heightened interest in their potential effects on the soil microbiome. The work presented is of significant interest and high quality, and I recommend minor revisions prior to publication:

-        Review the italics for gene names and bacterial genera (see "16S rRNA" from line 340 onward, "B. parachinensis" at line 414, and "Methylobacillus" at line 365).

-        Capitalize each keyword and separate them with a semicolon.

-        Line 60: “Estimates suggest that 43–63 and 30–44 million tons of MPs are introduced annually to European and North American farmlands, respectively.”

-        Lines 54 to 56: Reference Ibañez et al. 2021 (https://doi.org/10.3390/microorganisms9081619).

-        Lines 138 and 139: Use the plural form, "groups."

-        Line 399: Replace “indoor” with “greenhouse pot.”

-        Lines 411 and 412: Use the plural form, "groups."

-        Line 450: Replace “bipartite” with “paired.”

-        Once again, congratulations on the quality of the work presented.

Comments on the Quality of English Language

English is fine

Reviewer 3 Report

Comments and Suggestions for Authors

Dear authors, i read with interest the article  Structural and functional responses of rhizosphere bacteria to biodegradable microplastics in the presence of biofertilizers.

I think that the article is interesting but soime changes are required, specifically on the effects on bioplastic also on the growth of the plants. Results and discussion are very good. please modify the stastical analysis and specifiy the choose of the concentrations.

Specific comment are listed in the attached PDF. 

I will read again the modify version since I think that the topic is interesting

Round 2

Reviewer 1 Report

Comments and Suggestions for Authors

I thank the authors for addressing all of my comments. I don't have any new comments.

Reviewer 3 Report

Comments and Suggestions for Authors

The authors have implemented the ms as requested. I suggest the publication of it